# The Antitumor Effect of Lipophilic Bisphosphonate BPH1222 in Melanoma Models: The Role of the PI3K/Akt Pathway and the Small G Protein Rheb

**DOI:** 10.3390/ijms20194917

**Published:** 2019-10-03

**Authors:** Dominika Rittler, Marcell Baranyi, Eszter Molnár, Tamás Garay, István Jalsovszky, Imre Károly Varga, Luca Hegedűs, Clemens Aigner, József Tóvári, József Tímár, Balázs Hegedűs

**Affiliations:** 1Department of Pathology, Semmelweis University, H-1091 Budapest, Hungary; rittlerdomi@gmail.com (D.R.); baranyi2marci@gmail.com (M.B.); m.molnareszter@gmail.com (E.M.); garayt@gmail.com (T.G.); jtimar@gmail.com (J.T.); 2Pázmány Péter Catholic University, Faculty of Information Technology and Bionics, H-1083 Budapest, Hungary; 3Oncology Center, Semmelweis University, H-1091 Budapest, Hungary; 4Eötvös Loránd University, Faculty of Science, Institute of Chemistry, Department of Organic Chemistry; H-1117 Budapest, Hungary; jalso@caesar.elte.hu (I.J.); vukracs@caesar.elte.hu (I.K.V.); 5Department of Thoracic Surgery, Ruhrlandklinik, University Duisburg-Essen, D-45239 Essen, Germany; luca.hegedues@rlk.uk-essen.de (L.H.); clemens.aigner@ruhrlandklinik.uk-essen.de (C.A.); 6Department of Experimental Pharmacology, National Institute of Oncology, H-1122 Budapest, Hungary; jtovari@yahoo.com

**Keywords:** melanoma, lipophilic bisphosphonate, prenylation inhibition, Rheb

## Abstract

Malignant melanoma is one of the most metastatic cancer types, and despite recent success with novel treatment strategies, there is still a group of patients who do not respond to any therapies. Earlier, the prenylation inhibitor hydrophilic bisphosphonate zoledronic acid (ZA) was found to inhibit melanoma growth in vitro, but only a weaker effect was observed in vivo due to its hydrophilic properties. Recently, lipophilic bisphosphonates (such as BPH1222) were developed. Accordingly, for the first time, we compared the effect of BPH1222 to ZA in eight melanoma lines using viability, cell-cycle, clonogenic and spheroid assays, videomicroscopy, immunoblot, and xenograft experiments. Based on 2D and spheroid assays, the majority of cell lines were more sensitive to BPH. The activation of Akt and S6 proteins, but not Erk, was inhibited by BPH. Additionally, BPH had a stronger apoptotic effect than ZA, and the changes of Rheb showed a correlation with apoptosis. In vitro, only M24met cells were more sensitive to ZA than to BPH; however, in vivo growth of M24met was inhibited more strongly by BPH. Here, we present that lipophilic BPH is more effective on melanoma cells than ZA and identify the PI3K pathway, particularly Rheb as an important mediator of growth inhibition.

## 1. Introduction

Malignant melanoma is the most aggressive skin cancer with high mortality [1,2]. Earlier metastatic melanoma treatment was limited to aggressive chemotherapy and surgical resection. Currently, treatment options and survival have considerably improved through the approval of targeted and later immune checkpoint inhibitor therapies. Mutations of key proteins in the RAS/RAF/MEK/ERK and PI3K/Akt/mTOR signaling cascades were described in malignant melanoma including BRAF (35–60%), NRAS (15–30%), and PTEN (8–15%) [3,4,5]. Most of the melanoma patients with the BRAF mutation carry the BRAF V600E mutation, and these patients can be treated with BRAF-targeted therapy (e.g., vemurafenib, dabrafenib) in combination with MEK inhibitors (e.g., trametinib, cobimetinib) [3]. However, the development of acquired resistance against these inhibitors is still an unsolved problem [4]. PTEN null mutation has been found often concurrent with BRAF mutation, and it has been described as one of the contributing factor to BRAF inhibitor resistance [3,6]. There is still no promising targeted therapy against the mutant NRAS, despite the rapid development of RAS targeting research [7]. However, in certain cases, MEK inhibition triggered a response in patients with NRAS mutant melanoma [8,9]. Besides targeted therapy, immunotherapy has also proven to be an effective therapeutic option in malignant melanoma including immune checkpoint inhibitors such as the anti-CTLA-4 antibody ipilimumab or anti-PD1 antibodies pembrolizumab or nivolumab [4,10]. Despite all these promising therapy options, there is still a subset of patients without satisfying response to any of these treatments [8,10,11]. Therefore, the development of new treatment strategies is still necessary.

Nitrogen-containing bisphosphonates interrupt the mevalonate pathway in human cells by blocking farnesyl pyrophosphate synthase (FPPS), a key enzyme in this pathway. As a result, a posttranslational modification of proteins called prenylation is inhibited in the cells. This strongly affects small GTPases (e.g., Ras, Rho, Rac, Rheb), as a lack of prenylation prevents the essential membrane localization of these proteins [12,13]. Among these, Ras is part of the RAS/RAF/MEK/ERK pathway, while Rheb is a key protein in the activation of the mTOR complex in the PI3K/Akt/mTOR pathway. Therefore, nitrogen-containing bisphosphonates might be able to decrease the p-Akt, p-Erk, p-MEK, and p-mTOR levels and increase the expression of PTEN protein [14,15,16,17].

Hydrophilic amino bisphosphonates have been developed for the treatment of bone diseases, such as Paget’s disease and osteoporosis [18,19], and later, they were also applied against skeletal-related events and skeletal metastasis [20,21,22]. Recently, their direct antitumor effect was also demonstrated in human tumor cells, such as myeloma, breast cancer, prostate cancer, and melanoma cells, by the inhibition of migration, proliferation, and apoptosis induction [14,23,24,25]. Zoledronic acid (ZA) is one of the most potent amino-bisphosphonates. The in vitro antitumor activity of ZA was described in several tumor types including human melanoma cell lines [15,26,27,28]. However, in vivo antitumor application of hydrophilic bisphosphonates is limited because of their poor bioavailability and high affinity to bone mineral [29,30]. Therefore, several lipophilic variants of bisphosphonates were tested recently, and BPH1222 (BPH) was found to be the most potent analogue of ZA. The addition of a lipophilic side chain to the ZA skeleton in BPH enhanced its plasma half time and decreased its affinity towards the bone mineral compared to ZA [31,32,33]. Similarly to hydrophilic bisphosphonates, lipophilic bisphosphonates showed an antitumor effect on breast cancer, lung cancer, and glioblastoma cell lines [31,32,33]. Nevertheless, there are no data regarding lipophilic bisphosphonates in melanoma. Accordingly, in our study, we compared the effect of the conventional hydrophilic ZA to the lipophilic BPH on human melanoma cell lines in both in vitro and in vivo tumor models.

## 2. Results

### 2.1. BPH Decreased More Effectively Both Short-Term Cell Viability and Clonogenic Potential in Most Cell Lines than ZA

Short-term cell viability was assessed by the sulforhodamine B (SRB) assay after treatment with ZA or BPH for 72 h. We found that most of the cell lines were more sensitive to the lipophilic BPH than to the hydrophilic ZA (Figure 1). This effect was most pronounced in the BRAF mutation-harboring cell lines (Figure 1A, Appendix A), but was also present in the BRAF + PTEN mutant and in the BRAF/PTEN/NRAS wild-type cells (Figure 1B,D). However, in the case of NRAS mutant cell lines, ZA showed a higher growth inhibitory effect, especially on M24met cells (Appendix A, Figure 1C). The long-term effect of ZA and BPH treatments was also analyzed by a 10 day-long colony formation assay. BPH was also more effective in these settings when compared to ZA at inhibiting cell growth in most of the cell lines (Figure 2A). In the case of the BRAF mutant A375, WM35, A2058, and the wild-type VM47 cell lines, BPH inhibited colony formation much more effectively than ZA. In the WM3060, MEWO, and WM239 lines, the efficacy of the inhibitors was similar, and significant growth inhibition could be observed only at the higher 2 μM treatment with both inhibitors. Interestingly, the clonogenic potential of the M24met cell line was explicitly reduced by both treatments; however, the effect of ZA was slightly stronger. We compared the efficacy of BPH to ZA by calculating the ZA/BPH growth inhibition ratio for both the cell viability and for the clonogenic assay (Figure 1E and Figure 2B). We found that in the cell viability assay, the ZA/BPH ratio was the highest in the case of the BRAF mutant cell lines, especially at the higher treatment concentrations. Of note, at 25 µM treatment, BPH was over 26 fold more effective than ZA. Furthermore, the BRAF + PTEN mutant and BRAF/PTEN/NRAS wild-type cell lines were 2–3 fold more sensitive to BPH than to ZA (Appendix A, Figure 1E). In the colony formation assay, BPH had a 10-fold stronger effect than ZA in the BRAF mutant cell lines after 2-µM treatment (Figure 2B).

### 2.2. Cell Cycle Distribution and Apoptosis Induction upon Treatment with the Bisphosphonates

The distribution of the cells in the cell cycle phases was determined after treatment with both inhibitors (Appendix A). The ratio of the cell in the G0/G1 phase was decreased by the treatment in the A2058, WM239, and M24met cell lines. Additionally, moderate S phase arrest was observed in most of the cell lines after treatment with either both or one of the inhibitors, except for the M24met and VM47 cell lines. Regarding the subG1 phase, the highest increase was observed in the A375, M24met, and VM47 cell lines (Figure 3A). Furthermore, we also investigated the apoptosis induction via Western blot by cleaved-PARP/PARP protein detection (Figure 3B). We found that BPH was able to induce apoptosis, especially in the case of the BRAF and BRAF + PTEN mutant cell lines. However, ZA had a stronger apoptotic effect on the NRAS mutant M24met cell line than BPH. These results strongly correlated with the viability assay results (Figure 1).

### 2.3. Effect of the Prenylation Inhibitors on Erk, Akt, S6, and Rheb Activation

We used Western blot analysis to assess the effect of ZA and BPH treatment on protein activation in the RAS-RAF-MEK-ERK and RAS-PI3K-Akt-mTOR pathways (Figure 3B). We found that p-Erk expression was either unaffected or even increased upon treatment with both inhibitors in all cell lines, except in the M24met cell line, where a moderate decrease of p-Erk was observed. In contrast, the activation of Akt and S6 proteins was decreased by both inhibitors, but was more pronounced after BPH treatment. A375, VM47, and M24met cell lines were the most sensitive to BPH; in these cells the level of both p-S6 and p-Akt profoundly decreased. Interestingly, in the M24met cell line, ZA had a similar effect. In WM35 cells, the p-Akt level was reduced by both drugs; however, the level of activated S6 protein decreased moderately only after BPH treatment. In the case of WM239, A2058, and MEWO cells, the effect of both inhibitors on S6 and Akt proteins was not substantial. In WM3060, as the least sensitive cell line to BPH, Akt activation was even enhanced. However, S6 activation was unaffected after treatment with BPH. Furthermore, we tested the expression of another member of the PI3K pathway, the small G protein Rheb. We observed an electrophoretic mobility shift of Rheb in the A375, WM35, and VM47 cell lines upon treatment with BPH, which represents the unprenylated (upper line) and prenylated (lower line) protein [17,34,35]. Importantly, the strongest effect on Akt and S6 activation was found specifically in those cell lines where alterations of Rheb were observed. In addition, the protein level of Rheb was decreased after both BPH and ZA treatment in the M24met cell line. These results suggest that BPH has a stronger inhibitory effect on the PI3K pathway than on the RAS-RAF pathway and indicate that the Rheb protein may play a key role in the mechanism of action.

### 2.4. Efficacy of BPH to Inhibit Cell Migration Depends on the Baseline Motility of the Cells

Melanoma cells are usually highly motile [36]. In order to investigate the effect of BPH and ZA on the migratory capacity of melanoma cell lines, we measured the average migrated distance of the cells after treatment with ZA and BPH via time-lapse video microscopy (Figure 4, Appendix A). According to their baseline migration rate, we divided the eight melanoma cell lines into two groups: faster and slower cell lines (the cut-off value was 50 µm displacement in 18 h). We found that BPH inhibited the cell migration in three out of four faster cell lines, and this effect was statistically significant in two of them, namely in A2058 and VM47 cells. Additionally, the motility of the cell lines with lower migration potential did not change significantly upon treatment with the inhibitors.

### 2.5. Efficacy of ZA and BPH on Three-Dimensional Spheroid and In Vivo Xenograft Models

BPH1222 was expected to have an enhanced antitumor effect in 3D investigations due to its lipophilic character and better diffusion properties. We found that four out of the eight cell lines (A375, A2058, M24met, and VM47) were capable of forming 3D spheroids. These were treated with 2 or 5 µM ZA or BPH for 6–12 days (Figure 5). In the case of A375 cells, the effect of BPH was so strong that after six days of treatment, the spheroids completely dissociated, and the experiment had to be stopped. Furthermore, BPH also significantly inhibited the spheroid growth of A2058 and VM47 cells (Figure 5B,D,E). In contrast, in the M24met cells, ZA decreased spheroid formation more effectively than BPH, in good accordance with the 2D growth inhibition results (Figure 5C,E). All in vitro assay results and immunoblot experiments are summarized in Appendix A.

We also investigated the effect of ZA and BPH on the in vivo tumorigenicity of melanoma cells. We injected M24met cells subcutaneously into SCID female mice, and after tumor formation, the animals were treated with ZA or BPH for 23 days. We chose the M24met cell line for the in vivo experiment, because it was the only cell line with higher sensitivity to ZA than BPH in all the in vitro growth inhibition assays. We expected that in vivo BPH would be more effective even in these cells because of its lipophilic character and thus better diffusion properties. In good accordance with our expectation, in vivo BPH significantly reduced tumor growth, while ZA had no effect (Figure 6A, Appendix A). Similarly, tumor mass was moderately, albeit not significantly, decreased only after treatment with BPH (Figure 6B).

## 3. Discussion

Malignant melanoma is an aggressive tumor, and there are still patient groups without effective therapy; therefore, novel strategies are warranted to increase the survival of advanced melanoma patients [3,6,7]. Bisphosphonates are potent inhibitors of prenylation and thus of certain small G proteins, and they were previously shown to be potential antitumor agents [27]. In addition, the limitations of the usage of bisphosphonates, due to their poor bioavailability, facilitated the development of bisphosphonates with a lipophilic character. Lipophilic bisphosphonates have shown higher efficacy than the original hydrophilic compounds on lung, breast, and brain cancer cell lines. BPH1222 was the most potent ZA analogue, and it showed a higher antitumor effect than ZA. [31,32,33]. In our study, we presented for the first time in melanoma that the lipophilic BPH1222 inhibited growth and migration more effectively than the hydrophilic ZA and induced apoptosis in the majority of melanoma cell lines.

In line with our results, ZA was shown to have an inhibitory effect on melanoma cell lines in a dose-dependent manner in 2D experiments [26,27,28]. Furthermore, our previous results showed a mutation-dependent effect of ZA, namely BRAF mutant cell lines were less sensitive to ZA when compared to the BRAF + PTEN mutant cells. Interestingly, our data suggested that BPH was more effective than ZA on the cell lines with mutations in BRAF and in PTEN. Additionally, most of the melanoma cells were more sensitive to BPH than ZA both in short- and long-term viability assays. Similar observations were described in lung cancer by Xia et al. [31]. Interestingly, the NRAS mutant M24met melanoma cell line was more sensitive to ZA than BPH. Of note, ZA was described to inhibit EGFR at a lower concentration than BPH [37,38], and EGFR was overexpressed in M24met cells [39]. Consequently, ZA might exert effects on M24met cells not only as a prenylation inhibitor, but also as an EGFR inhibitor, which might explain the enhanced sensitivity of the M24met cell line to ZA in vitro.

Additionally, previous studies on cell cycle analysis revealed that ZA caused an accumulation of the cells in the S-phase [27]. We also detected S-phase arrest in a number of our cell lines after bisphosphonate treatment. In addition, in four cell lines (A375, WM35, VM47, and M24met), the ratio of the cells increased in the subG1 phase after treatment with BPH, suggesting induction of cell death.

It was previously shown that p-Akt and p-Erk levels can be altered by treatment with ZA at higher concentrations [14,16,26]. We found that Erk activation was not influenced by either inhibitor except for the M24met cell line, where modest reduction was detected. In line with previous observations, we found that ZA was able to decrease the activation of the PI3K pathway (Akt, S6) only in the M24met cell line. However, the lipophilic BPH reduced the p-S6 and p-Akt level pronouncedly in four cell lines (A375, WM35, M24met, and VM47), as was already described in lung cancer cells by Xia et al. [31].

Rheb is a small G-protein that also undergoes prenylation posttranslationally. We hypothesized that the p-S6 and p-Akt decrease in these four cell lines could be associated with prenylation inhibition of Rheb following BPH treatment. Our results showed that Rheb protein was separated into two bands by Western blot upon treatment with BPH in the case of the A375, WM35, and VM47 cell lines (Figure 3B). Previous studies identified the lower band as prenylated and the upper band as unprenylated forms of Rheb protein [17,34,35]. Furthermore, in the case of A375 and VM47, the prenylated form of the protein almost disappeared upon treatment with BPH. The prenylated Rheb was decreased in the WM35 cell line after treatment with BPH, and the unprenylated form appeared as well. Additionally, in the case of the M24met cell line, the total amount of Rheb decreased upon treatment with both BPH and ZA. Of note, these four cell lines were detected as the most sensitive cell lines to BPH, based on growth inhibition and apoptosis induction assays. These results altogether suggest that Rheb prenylation inhibition could be essential for the mode of action of BPH.

Interestingly, in the least-sensitive WM3060 cells, NRAS and Rac1 mutant cell line, an enhanced level of p-Akt was observed upon BPH treatment. Since Rac1 is also prenylated, its mutation might have an impact of the cellular effect of BPH. In line with our observation, Rac1 mutation was described previously to play a role in the insensitivity of the cells against RAF inhibitors [40].

Previous studies indicated that ZA is able to inhibit invasion of breast and prostate cancer cells [41,42]. However, ZA failed to decrease the 2D motility of the melanoma cell lines both in our earlier study and in the current investigation [26]. In contrast, BPH was able to reduce the melanoma motility, especially in faster migrating cell lines.

Since three-dimensional in vitro models are increasingly important in preclinical studies, the 3D spheroid growth assay was also performed [43]. In our study, a significant spheroid growth inhibitory effect was detected in the case of A2058, VM47, and M24met cell lines with 5-µM BPH (Figure 5), similarly to the long-term viability assay results (Figure 2A). However, in the case of the M24met cell line, ZA demonstrated a stronger effect when compared to BPH, in line with the 2D experiments. These 3D results had a strong correlation with the 2D growth inhibition findings.

Finally, we also performed a xenograft experiment to validate the effectiveness of BPH also in an in vivo setting, and ZA did not show a pronounced antitumor effect, in line with previous studies [26,44]. Importantly, BPH was able to decrease the volume and moderately the mass of the tumors significantly, similar to the previous in vivo results using a lung cancer cell line [31]. We confirmed, in line with previous observations, that the lipophilic profile of BPH could lead to growth inhibition since this compound has better bioavailability and less affinity to the bone mineral and consequently stays for a longer period of time in the circulation [31,32].

## 4. Material and Methods

### 4.1. Cell Lines and Reagents

Eight human melanoma cell lines were investigated in this study. Cells were divided into four groups based on their NRAS, BRAF, and PTEN mutational status (Table 1). A375, A2058, and MEWO lines were purchased from ATCC, while WM35, WM239, and WM3060 cell lines were from the Wistar Institute. The M24met cell line was kindly provided by Barbara M. Mueller (Scripps Research Institute, La Jolla, CA), established from an invaded lymph node of a nude mouse [39]. The VM47 cell line was established from a melanoma brain metastasis at the Institute of Cancer Research, Medical University of Vienna, Austria [45]. All cell lines were maintained at 37 °C and 5% CO_2_ in a humidified atmosphere in tissue culture flasks in DMEM with 4500 mg/dm^3^ glucose, pyruvate, and L-glutamine (Lonza, Basel, Switzerland) completed with 10% fetal bovine serum (FBS) (Gibco-BRL Life Technologies, Glasgow, U.K.) and 1% penicillin-streptomycin-amphotericin (Lonza). As the hydrophilic bisphosphonate, zoledronic acid (ZA) was purchased from Novartis (Basel, Switzerland), and lipophilic BPH1222 (BPH) was synthetized at the Department of Organic Chemistry, Eötvös Loránd University, Budapest (Appendix A).

### 4.2. Sulforhodamine B Assay

The short-term (72 h) efficacy of the inhibitors on melanoma cell lines was measured based on the protein content of the cells via the SRB assay. Briefly, melanoma cells were seeded in the inner 60 wells of a 96-well-plate at 1000–6000 cells/well density, according to the proliferation rate of the cell lines. The outer wells of the plate were filled with PBS (Lonza) to avoid the evaporation of the medium from the inner wells. Following the adherence of the cells (after 24 h of incubation), they were treated with increasing concentrations of ZA or BPH. After 72 h, cells were fixed with 10% trichloroacetic acid and stained with SRB dye for 15 min. The excess dye was washed out by 1% acetic acid solution. After drying, the bound dye was dissolved in 10 mM Tris-HCL buffer (pH = 7.4), and the optical density was measured at 570 nm using a micro plate reader (EL800, Bio-Tec Instruments, Winooski, VT, USA).

### 4.3. Clonogenic Assay

The long-term (10 days) effect of the inhibitors on the colony forming potential of the melanoma cell lines was assessed via a clonogenic assay. Briefly, melanoma cells were plated in a 6-well plate at 1000–4000 cells/well (except for WM3060: 10000 cells/well). Upon adhesion, the medium was replaced with fresh medium supplied with ZA or BPH in 1 or 2 µM concentrations, and this step was repeated every 3–4 days. In these long-term experiments with multiple treatment over time, lower concentrations were chosen based on the short-term viability assay results. After 10 days of treatment, cell monolayers were fixed by a methanol:acetic acid mixture (3:1) for 30 min. The protein content of the fixed colonies was stained by crystal violet dye for 30 min. Excess dye was removed by distilled water, and after drying, attached dye was dissolved in 2% SDS solution, while the optical density was determined at 5700 nm with the micro plate reader (EL800).

### 4.4. Time-Lapse Video Microscopy

Cell migration analysis was performed by video microscopy. Briefly, melanoma cell lines were seeded in the inner 12 wells of a 24-well plate and were kept 24 h to attach. Before the measurement was started, culture medium was changed to CO_2_ independent medium (Gibco) with 10% FBS and 4 mM glutamine. PBS was filled in the outer wells to reduce the evaporation from the inner wells. During the measurement, phase-contrast pictures were taken from three non-overlapping fields of the wells every 10 min at 37 °C and a normal atmosphere, in a custom-designed incubator. Before treatment, the baseline record was established for 24 h, and then, cells were treated with ZA or BPH (10 µM). Treatment concentration was defined considering the short-term viability assay results to make sure that we detected migration changes of the cells and, to a lesser extent, the impact of cell death on migration. Pictures were analyzed with manual marking of the living cells with a custom-made cell-tracking program. Individual cells (approximately 30 cells per treatment) were tracked in the first 24 h upon treatment. Net displacement (distance between the locations at the beginning and the end of each time interval) of the cells was determined by the average of at least two independent measurements.

### 4.5. Immunoblot Analysis

Changes in protein expression after treatment with ZA or BPH were analyzed via Western blot assay. Cell lines were seeded in a 6-well plate, 1–2 × 10^5^ cells/well. Upon attachment, fresh medium supplemented with ZA or BPH at 10-µM concentration was added for 48 h. The concentration was determined based on the short-term viability assay results. After treatment, in order to precipitate cellular protein, ice-cold 6% trichloroacetic acid was applied for at least 1 h at 4 °C. The supernatant medium was also collected and centrifuged, and ice-cold 6% TCA was added to the pellet, as well. Samples were centrifuged at 3500 rpm, 4 °C, for 15 min, then TCA was removed, and the pellets were dissolved in SSP buffer (containing 2% SDS, 10% glycerol, 62.5-mM Tris-HCl pH 6.8, 5-mM EDTA, 125 mg/mL of urea, 10 mM dithiothreitol, and 0.14 mg/mL of bromophenol blue). Total protein concentration was defined by the Qubit protein assay kit (Thermo Scientific, Waltham, MA, USA). Then, 25 µg of protein were loaded on 10% polyacrylamide gels, then separated by gel electrophoresis and transferred to PVDF membranes (Thermo Scientific). The following primary antibodies were used in a dilution of 1:1000 overnight at 4 °C: PARP, p-Akt/Akt, p-Erk1/2,/Erk1/2, p-S6/S6 and Rheb (Cell Signaling; #9542, #4058, #9272, #9101, #9102, #2215, #2217, #13879, respectively) and as the loading control, anti ß-tubulin (Abcam, ab6046). Then, secondary HRP-labelled anti-rabbit antibody (Jackson ImmunoResearch, West Grove, PA, USA) was applied in a dilution of 1:10,000 for 1 h at room temperature, and for visualization, Pierce ECL Western Blotting Substrate (Thermo Scientific) was used. The signal was detected by the development of CL-XPosure Film (Thermo Scientific). Occasionally, the G-Box Chemi XR5 system (Synoptics Group, Cambridge, U.K.) was used for signal detection.

### 4.6. Cell Cycle Analysis

The distribution of cells in the cell cycle phases was analyzed by the DNA content of the cells. Briefly, cells were plated in 6-well plates at 1–2 × 10^5^ cells/well and were treated with 10 µM ZA or BPH for 72 h. Then, supernatant and attached cells were collected, and lysis buffer with DAPI was added for 5 min at 37 °C. Afterwards, stabilization buffer was applied, and the samples were measured by NucleoCounter NC-3000™ system (Chemometec, Allerod, Denmark).

### 4.7. 3-Dimensional Spheroid Assay

Four out of the eight melanoma cell lines (A375, A2058, M24met, VM47) were suitable to create 3D spheroids via the hanging drop method. Briefly, 50–200 cells in 6 microliter drops were placed in the inner 60 circles of the lid of a 96-well plate. We also placed drops without cells on the outer circles of the lid, and the wells were filled with 100 µL of PBS to reduce evaporation of the cell-containing drops. After 4–7 days, the spheroids were transferred from the drops to Ultra Low Attachment (ULA) 96-well plates (Corning Incorporated, Corning, NY) filled with medium. When the spheroids sank down to the bottom of the wells, the 100 µL of supernatant medium were removed. Then, spheroids were treated every third day with 2 or 5 µM ZA or BPH for 6–12 days. Treatment concentrations were chosen considering both the short- and long-term viability assay results. Photos were taken using a 4× objective every third day. To analyze the changes of spheroid volume, the area and radius of the spheroids were measured via the ImageJ program, and the equivalent sphere volume was calculated (4/3 × π × radius^3^).

### 4.8. In Vivo Experiment

In order to compare the effect of ZA and BPH on the in vivo tumorigenicity of M24met cells, 1.5 × 10^6^ cells in 150 µL were injected subcutaneously into SCID female mice. When tumors became measurable (7 days), mice were divided into control (0.9% NaCl) and ZA and BPH treatment groups (10 mice each). ZA and BPH were administered intraperitoneally twice a week at a dose of 1.47vµmol/kg for 23 days. The diameter of the growing tumors was measured by a caliper, and tumor volumes were calculated with the formula for a prolate ellipsoid’s volume (4/3 × π × width^2^ × length). On the last day of the experiment, the size of the tumors was measured, then mice were sacrificed, and the tumor tissues were removed and weighed. All animal experiments were carried out in accordance with the Guidelines for Animal Experiments and were approved for the Department of Experimental Pharmacology in the National Institute of Oncology, Budapest, Hungary (Permission Number PEI/001/2574–6/2015, 12/10/2015)

### 4.9. Statistics

A two-way ANOVA and non-parametric Kruskal–Wallis test were used, followed by Bonferroni’s and Dunn’s post hoc test, respectively, to determinate the statistical difference between groups on 2D viability assays. One-way ANOVA and Dunnett’s or Tukey’s post hoc test were applied on migration and cell cycle analysis, respectively. One-way repeated measures ANOVA was used followed by Tukey’s post hoc test on 3D and in vivo experiments to compare treated groups. Statistical significance was indicated as * *p* < 0.05, ** *p* < 0.01 and *** *p* < 0.001. All statistical analyses were performed in GraphPad Prism 5 (GraphPad Software Inc, San Diego, CA, USA).

## 5. Conclusions

In conclusion, BPH1222 was proven to be a more effective antitumor agent than ZA on most of the melanoma cell lines in in vitro and in vivo experiments. Furthermore, our study suggested that the antitumor effect of BPH might be mediated through the PI3K pathway due to its Rheb prenylation inhibitory effect. Based on our findings, further investigations are warranted to explore whether lipophilic bisphosphonates have a therapeutic potential in combinatorial treatments against malignant melanoma.

## Figures and Tables

**Figure 1 ijms-20-04917-f001:**
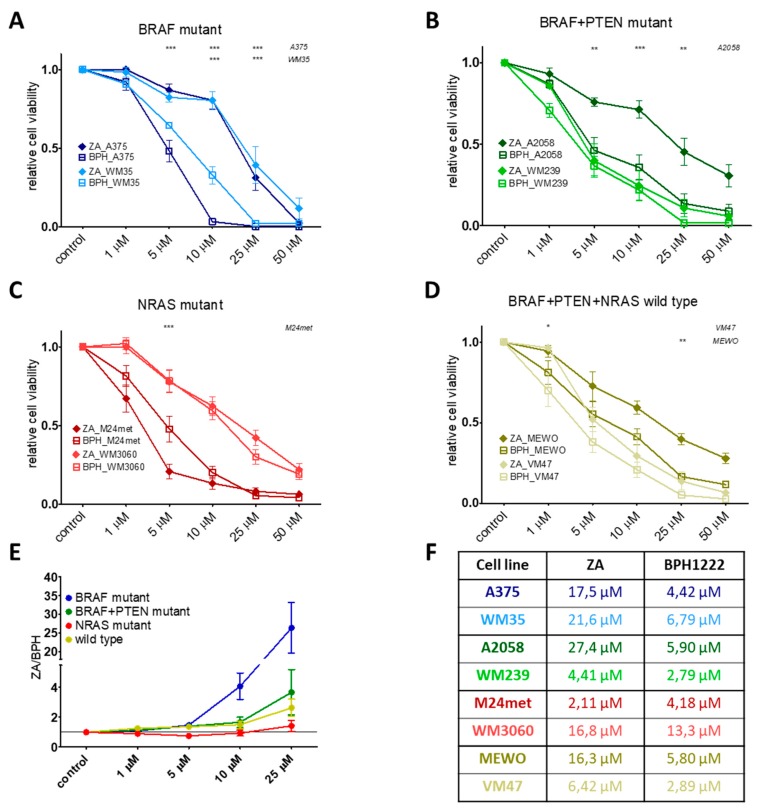
Short-term (72 h) sulforhodamine B (SRB) assay results on melanoma cell lines after treatment with different concentrations of zoledronic acid (ZA) or bisphosphonate (BPH). (**A**) BRAF mutant A375 and WM35, (**B**) BRAF and PTEN mutant A2058 and WM239 (**C**) NRAS mutant M24met and WM3060, and (**D**) BRAF, PTEN, and NRAS wild-type MEWO and VM47 cell lines. All lines were more sensitive to BPH than ZA, except for the NRAS mutant M24met cell line, where ZA was more effective. Data are shown as the mean ± SEM from at least three independent measurements. Asterisks mean a significant difference between BPH and ZA by * *p* < 0.05, ** *p* < 0.01, and *** *p* < 0.001. (**E**) The graph shows the efficacy of BPH compared to ZA by the ZA/BPH ratio of the averages. Results are from the short-term (72 h) viability assays, as one point is the average of all of the given mutation group viability data at the indicated concentration. Data are shown as the mean ± SEM from at least eight independent experiments. (**F**) IC_50_ values upon treatment with ZA or BPH for 72 h.

**Figure 2 ijms-20-04917-f002:**
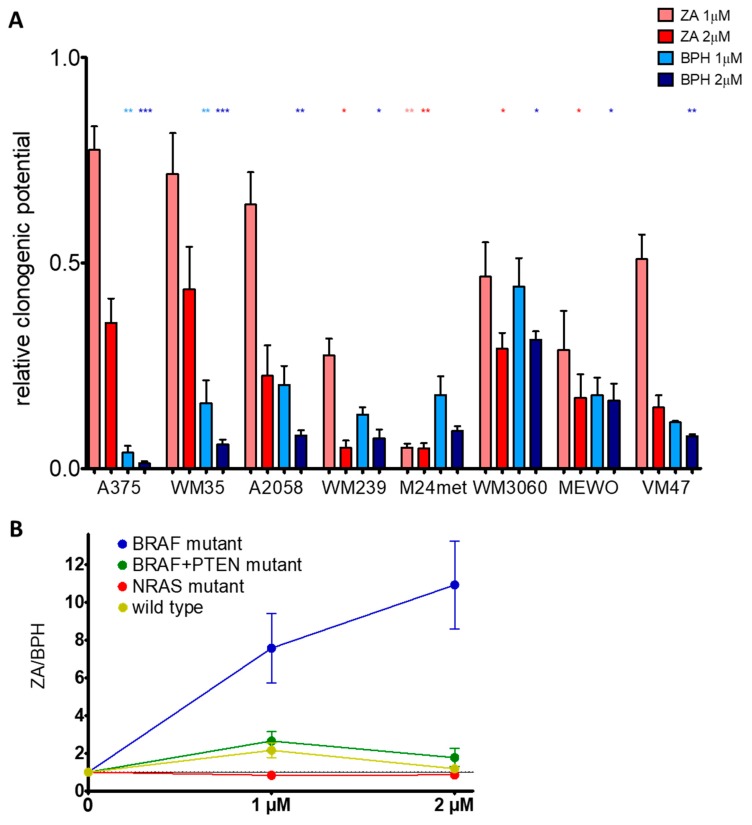
(**A**) Long-term (10 days) effect of the inhibitors on the colony-forming potential of the melanoma cell lines. Most of the cell lines were more sensitive to BPH, except for the M24met cell line. Data are shown as relative to the control and the average of at least three independent measures ± SEM. Asterisks mean a significant difference between the control and BPH (blue star) or ZA (red star) by * *p* < 0.05, ** *p* < 0.01, and *** *p* < 0.001. (**B**) The graph demonstrates the efficacy of BPH compared to ZA by the ZA/BPH ratio. Results are from long-term (10 days) clonogenic assays, as one point is the average of all of the given mutation group viability data at the indicated concentration. Data are shown as the mean ± SEM from at least eight independent experiments.

**Figure 3 ijms-20-04917-f003:**
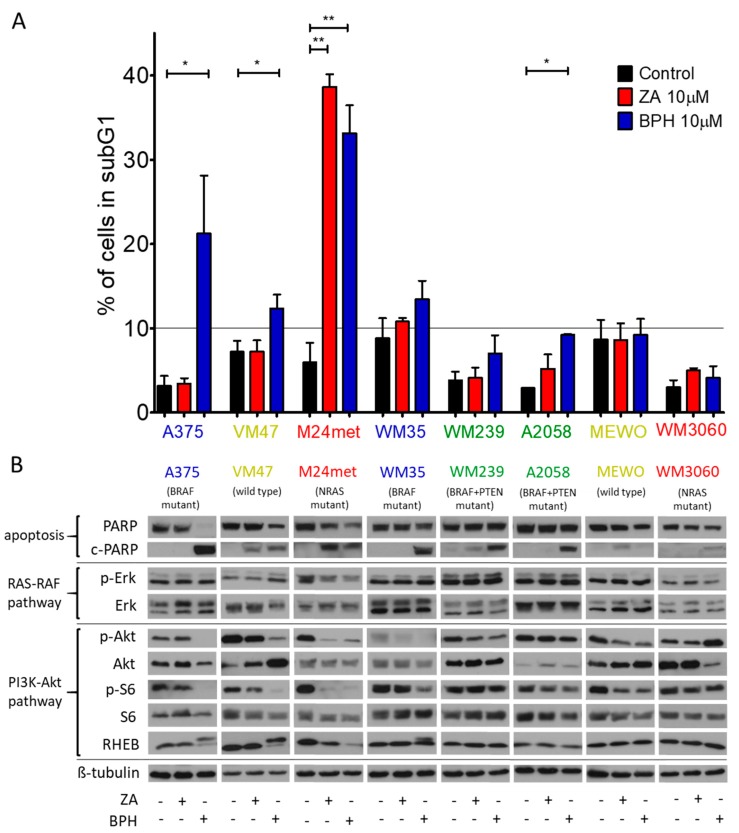
(**A**) The cell cycle distribution was examined after 72 h of treatment with 10 µM ZA or BPH. In most cell lines, BPH increased more strongly the ratio of cells in the subG1 phase except for the M24met cell line. Data are shown as the average ± SD from two or three measurements. Asterisks mean a significant difference between the control and treated groups by * *p* < 0.05 and ** *p* < 0.01. (**B**) Western blot analysis was performed to detect the apoptosis induction and protein activation after 48 h-long treatment with 10 µM ZA or BPH. C-PARP was detected in most of the cell lines, particularly after treatment with BPH. Levels of total and phosphorylated Akt, S6, and Erk, as well as Rheb and c-PARP/PARP protein were analyzed. β-tubulin was used as the loading control. In the majority of the cell lines, activation of S6 and/or Akt decreased especially after BPH treatment, while Erk activation did not change substantially. The Rheb protein level was modified by BPH treatment in four cell lines (A375, WM35, VM47, M24met). Immunoblots are representative images from at least three independent measurements. The colors of the cell names represent the mutational group of the cells as BRAF (blue), BRAF + PTEN (green), NRAS mutant (red), and BRAF + PTEN + NRAS wild-type (yellow).

**Figure 4 ijms-20-04917-f004:**
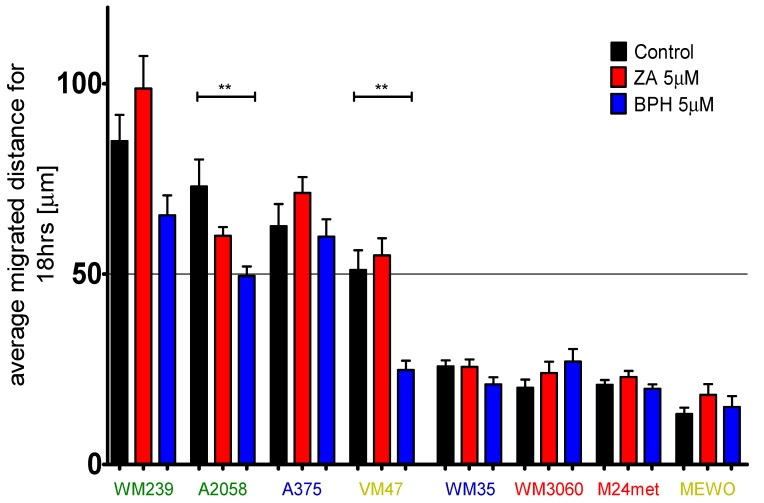
Average migrated distance of melanoma cell lines upon treatment with 5 µM ZA or BPH was measured by time-lapse video microscopy. Cell lines were divided into two groups according to their baseline migratory capacity. The faster lines were more sensitive to the treatment with BPH than ZA. Moreover, ZA could cause an adverse effect on migration. Slower lines were insensitive to the treatment. Blue, green, red, and yellow colors indicate the mutation status of the cells, as the BRAF mutant, BRAF + PTEN mutant, NRAS mutant, and BRAF + PTEN + NRAS wild-type, respectively. Data are shown as the average of the migrated distance for 18 h ± SEM from at least two independent measurements and three non-overlapping microscopic fields. Asterisks mean a significant difference between control and BPH by ** *p* < 0.01.

**Figure 5 ijms-20-04917-f005:**
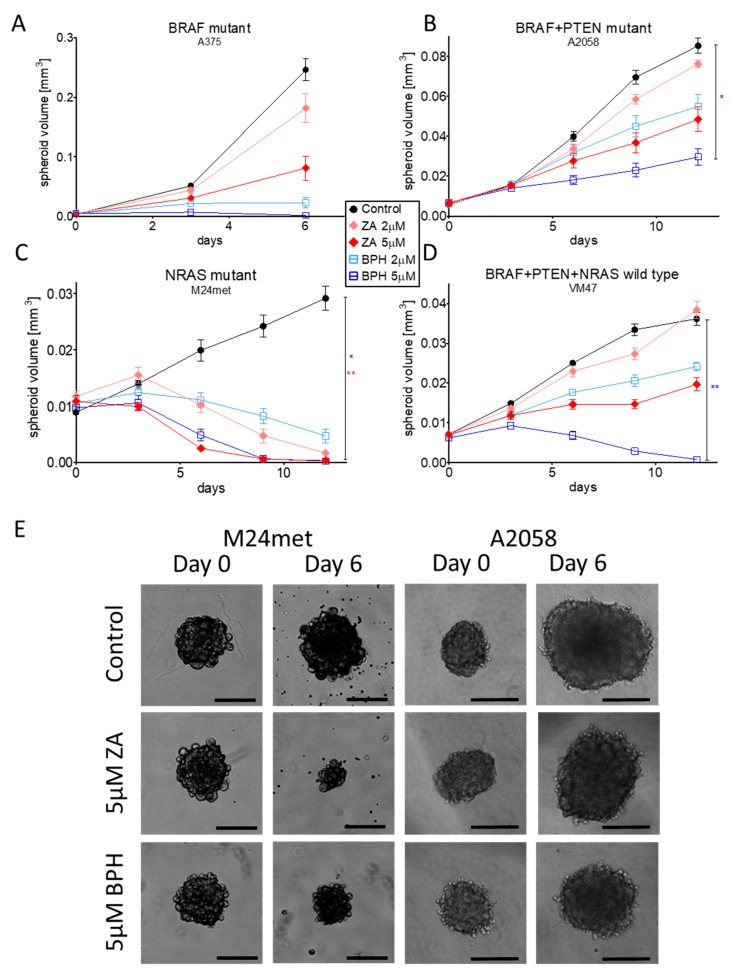
Three-dimensional spheroid growth experiment was performed to compare the effect of ZA and BPH. In the case of A375, A2058, and VM47 cell lines (**A**,**B**,**D**), BPH had a stronger effect compared to ZA, while the M24met cell line was more sensitive to ZA (**C**). Data are shown as the average ± SEM from at least two independent experiments. Asterisks mean a significant difference between the control and BPH (blue star) or ZA (red star) by * *p* < 0.05 and ** *p* < 0.01. (**E**) Representative pictures show the effect of the inhibitors on the sixth day after treatment with 5-µM ZA or BPH. Scale bar means 200 µm.

**Figure 6 ijms-20-04917-f006:**
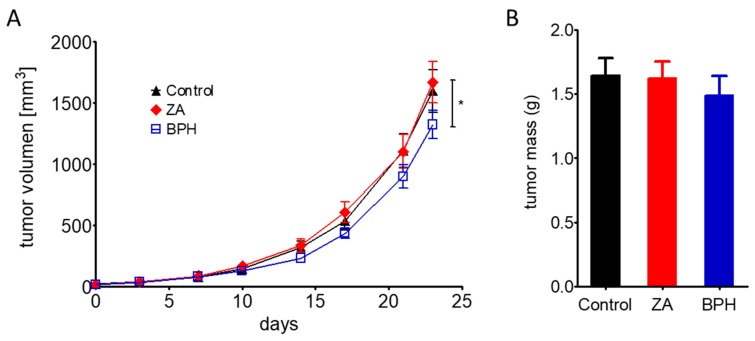
Antitumor effect of ZA (red) compared to BPH (blue) on the M24met cell line in SCID female mice. (**A**) Results after 23 day-long treatment with 1.47-µmol/kg ZA or BPH showed that BPH was significantly more effective at inhibiting tumor growth than ZA. (**B**). BPH slightly decreased the tumor mass; however, it was not statistically significant. Data are shown as the average ± SEM from *n* = 10 groups. Asterisks mean a significant difference between BPH and ZA by * *p* < 0.05.

**Table 1 ijms-20-04917-t001:** Melanoma cell lines by NRAS, BRAF, and PTEN mutational status and karyotype. (n.a = not applicable).

Cell Line	BRAF	PTEN	NRAS	Karyotype	Reference
A375	V600E	wt	wt	hypo triploid (62)	[46,47]
WM35	V600E	wt	wt	near-triploid (68)	[26,48,49]
A2058	V600E	null	wt	near-diploid (43-48)	[46,50]
WM239	V600D	null	wt	near-diploid (43)	[26,51,52]
M24met	wt	wt	Q61R	hyper diploid (53-57)	[26,39]
WM3060	wt	wt	Q61K	n.a.	[26,53]
MEWO	wt	wt	wt	near-diploid (43-46)	[46,54]
VM47	wt	wt	wt	n.a.	[26,55]

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
