# Peer review of "The Antitumor Effect of Lipophilic Bisphosphonate BPH1222 in Melanoma Models: The Role of the PI3K/Akt Pathway and the Small G Protein Rheb"

_ijms, 2019, doi:10.3390/ijms20194917_

Round 1

Reviewer 1 Report

In summary the paper shows that the lipophilic forrm or lipophilic bisphosphonate

3 BPH1222 are more appropriate to kill melanoma cells.

In the abstract, 3D sould be replaced by « spheroid assay ».

Figure 1 : The panels are hardly readable as presentend. I suggest to replace curve symbols by different colors for each cell line.

In figure 1 E, no error bar is provided.

In figure 2 B, no error bar is provided.

Figure 3 : why are the names of the cell lines presented in different colors, does this mean something ? the western blot panels are quite difficult to figure out. To me,

other options of presentation should be chosen. Perhaps, analysis concerning each of the pathways should be separated for more clarity.

Although, a table of the description of the cell lines use is included in the manuscript, the caryotype (and /or fish caryotype) analyses are not provided. Usually, these cell lines exhibit chomosomal aberrations (number of chromosomes, the structure an the many mutations they bear in addition to those analysed and characterized mutations). In the context of our knowwledge today it seems important to consider these parameters before any toxycologic analysis.

That the cytotoxic potential of a lipophilic substance exhibits higher killing properties in not a real surprise. The presented findings should be accompanied by animal studies ; these kind of studies are evoked in the discussion but their outcome is not clear in the context of this manuscript : « BPH was able to moderately decrease the volume and mass of the tumors, similar to « the later previous in vivo results using a lung cancer cell line [31]. We hypothesize that the lipophilic profile of 281 BPH could lead to growth inhibition since this compound has better bioavailability and less affinity 282 to the bone mineral and consequently stays for a longer period of time in the circulation [31,32] ».

Reviewer 2 Report

Thank you for submitting the manuscript to the IJMS. Authors showed the effect of 2 bisphosphonate agents on melanoma and the difference between the 2 agents with 8 melanoma cell lines in vitro and in vivo. The manuscript is well organized and very informative, however, there are many concerns before accepting. Please read and consider the concerns as follows:

1. Figure 1 shows the dose dependency of the 2 bisphosphonate agents. Since authors chose 5 different concentrations of the agents and the dose response curves were pretty good shapes, IC50 values can be calculated and should be shown.

2. In this article, authors are focusing on a novel treatment strategies for malignant melanoma. Since several anti-tumor drugs have already applied for clinical use, at least one of the existing anti-tumor agents should be compared with BPH1222 in Figure 1 2, 5 or 6.

3. In Figure 2, the concentration of the bisphoshonates was 1 and 2 micro-moler, whereas 10 micro-moler was chosen in Figure 3, as well as 5 micro-moler was chosen in Figure 4 and 5. Why these different concentrations were used? Please explain the reason why the different concentrations were used.

4.In line 183, authors describe "at least 2 independent measurements", however, in line 329, authors also describe "average of 3 independent measurements". There is an inconsistency between them. Please correct them.

5. In Figure 4, the average migrated distance of the cells for 18hrs was shown. How this distance was measured? For example, the distance between the starting point and the ending point, or the total distance of traces of the migrated cells, etc... Also, how many cells were traced and analyzed? Please explain.

6. In Figure 4, only the average migrated distance was shown. Since authors performed Time-lapse video microscopy, time lapse images or movies should be shown as well.

7. In line 124, authors describe the highest increase was observed in A375, WM35, M24met and VM47 cell lines. However, in Figure 3A, the WM35 cells do not have statistically significant difference between control and BPH. WM35 should be removed from the sentence in line 124.

8. Figure 4, there are only 2 cell line with BPH showed the statistical difference (A2058/VM47 with BPH). However, authors describe BPH inhibited the migration of 3 out of 4 faster cell lines. Why?

9. In line 170, authors describe ZA treatment decreased the migration of A2058, however, there was no statistical significant. This sentence should be removed.

10. In line 171, authors describe ZA treatment even slightly increased cell migration, however, there was no statistical significant. This sentence should be removed.

11. In 173-175, authors describe the average migrated distance was slightly increased.... 20% by BPH, , however, there was no statistical significant. This sentence should be removed.

12. The word "data" is plural. There are many mistakes of the verb for "data".

13. The level of English description is not satisfying. The manuscript needs proof reading by a native Enflish speaker.

Reviewer 3 Report

In the original manuscript entitled “The antitumor effect of lipophilic bisphosphonate BPH1222 in melanoma models: the role of PI3K/Akt pathway and the small G protein Rheb”, Rittler and colleagues compared the effect of BPH1222 to ZA in eight melanoma lines using viability, cell-cycle, clonogenic and spheroid assays, videomicroscopy, immunoblot and xenograft experiments. Overall the finding is novel, a major concern is raised for the xenograft assay: why the authors used 1.47 μmol/kg ZA or BPH for the test? The tumor mass difference did not reach statistical significance, so the authors should not conclude “tumor mass was decreased”. The authors should try other concentration of drugs to see if they have significant effect, the authors should also show the picture of xenograft tumors.

Round 2

Reviewer 1 Report

The manuscript has bee substantially improved according according to referees comments. 

Author Response

The manuscript has been substantially improved according to referees’ comments.
Response: Thank you for your review.

Reviewer 2 Report

To Authors,

Thank you for responding to each points. Some responses are OK but others are still not satisfying as below. Thank you for re-considering them.

1, 2, 4, 5, 7, 9, 10, 1, 12, and 13 : OK

3 and 8:

Thank you for explaining to the concerns. Please explain them to readers in the manuscript.

6:

Thank you for explaining and adding images to figure S4. The method of measuring the distance is now clear. However, in these images, it is unclear which cell in hour 0 to migrated to which cell in hour 24. Also, in images in hour 24, more cells are seen than cells in hour 0. Did these cells in hour 24 migrated from other part, or divided from the cells in hour 0? In this kind of figure, only showing the images is not enough to explain the migration data. It is better to indicate the same cell in both time points. Merged images in different colors may be useful to show this.

Also the images need a scale bar.

Author Response

Dear Editor and Referees,
We would like to thank the Editor and the Reviewers for the additional suggestions and remarks. The manuscript has now been revised and extended following the suggestions. All concerns raised by the reviewers are addressed in the point-by-point response below. Newly added sections are highlighted in the manuscript text.
We hope that our revised manuscript is now suitable for publication in International Journal of Molecular Sciences.

Reviewer 2 comments:
Thank you for responding to each points. Some responses are OK but others are still not satisfying as below. Thank you for re-considering them.
1, 2, 4, 5, 7, 9, 10, 1, 12, and 13 : OK
3 and 8:
Thank you for explaining to the concerns. Please explain them to readers in the manuscript.
Response: We inserted the explanation into the manuscript according to the suggestion. (Page 6 Line 179-181; Page 11 Line 326-327; Page 12 Line 340-342, 350-351, 381-382)
6:
Thank you for explaining and adding images to figure S4. The method of measuring the distance is now clear. However, in these images, it is unclear which cell in hour 0 to migrated to which cell in hour 24. Also, in images in hour 24, more cells are seen than cells in hour 0. Did these cells in hour 24 migrated from other part, or divided from the cells in hour 0? In this kind of figure, only showing the images is not enough to explain the migration data. It is better to indicate the same cell in both time points. Merged images in different colors may be useful to show this.
Also the images need a scale bar.
Response: We thank you for this important remark. Now we changed Figure S4 according to the suggestion. Cells have been marked with numbers on the picture of hour 0 and hour 24. Cells without numbers mean that those cells are not on both of the pictures, either they went out from the picture or they came from outside during this 24 hours. Divided cells have been marked with the same number and an ‘a’ or ‘b’ as daughter cells. Pictures on the third column show the merged images of hour 0 (red cells) and hour 24 (green cells). This additional explanation has been inserted to the description of Figure S4 as well. Scale bar has been inserted and means 40μm.

Reviewer 3 Report

The authors addressed my concern.

Author Response

The authors addressed my concern.
Response: Thank you for your review.

Round 3

Reviewer 2 Report

Dear Authors,

Thank you for considering the comments and for revising the manuscript. Now the manuscript is much better than the original version and is satisfying. I hope the study will be helpful for all readers.

Best regards.